# Characteristics of Children Likely to Have Spontaneous Resolution of Obstructive Sleep Apnea: Results from the Childhood Adenotonsillectomy Trial (CHAT)

**DOI:** 10.3390/children8110980

**Published:** 2021-10-29

**Authors:** Solveig Magnusdottir, Hugi Hilmisson, Roy J. E. M. Raymann, Manisha Witmans

**Affiliations:** 1MyCardio LLC, SleepImage, 3003 E 3rd Avenue, Denver, CO 80206, USA; hugi.hilmisson@sleepimage.com (H.H.); roy.raymann@sleepimage.com (R.J.E.M.R.); 2Department of Pediatrics, Faculty of Medicine & Dentistry, University of Alberta, Edmonton, AB T6G 2R3, Canada; manishawitmans@gmail.com

**Keywords:** sleep apnea, sleep quality, cardiopulmonary coupling, children

## Abstract

Objective: To evaluate if cardiopulmonary coupling (CPC) calculated sleep quality (SQI) may have a role in identifying children that may benefit from other intervention than early adenotonsillectomy (eAT) in management of obstructive sleep apnea (OSA). Methods: A secondary analysis of electrocardiogram-signals (ECG) and oxygen saturation-data (SpO_2_) collected during polysomnography-studies in the prospective multicenter Childhood Adenotonsillectomy Trial (CHAT) to calculate CPC-SQI and apnea hypopnea index (AHI) was executed. In the CHAT, children 5–9 years with OSA without prolonged oxyhemoglobin desaturations were randomly assigned to adenotonsillectomy (eAT) or watchful waiting with supportive care (WWSC). The primary outcomes were to document change in attention and executive function evaluated with the Developmental Neuropsychological Assessment (NEPSY). In our analysis, children in the WWSC-group with spontaneous resolution of OSA (AHI_Obstructive_ < 1.0) and high-sleep quality (SQI ≥ 75) after 7-months were compared with children that showed residual OSA. Results: Of the 227 children randomized to WWSC, 203 children had available data at both baseline and 7-month follow-up. The group that showed resolution of OSA at month 7 (*n* = 43, 21%) were significantly more likely to have high baseline SQI 79.96 [CI_95%_ 75.05, 84.86] vs. 72.44 [CI_95%_ 69.50, 75.39], *p* = 0.005, mild OSA AHI_Obstructive_ 4.01 [CI_95%_ 2.34, 5.68] vs. 6.52 [CI_95%_ 5.47, 7.57], *p*= 0.005, higher NEPSY-attention-executive function score 106.22 [CI_95%_ 101.67, 110.77] vs. 101.14 [CI_95%_ 98.58, 103.72], *p* = 0.038 and better quality of life according to parents 83.74 [CI_95%_ 78.95, 88.54] vs. 77.51 [74.49, 80.53], *p* = 0.015. The groups did not differ when clinically evaluated by Mallampati score, Friedman palate position or sleep related questionnaires. Conclusions: Children that showed resolution of OSA were more likely to have high-SQI and mild OSA, be healthy-weight and have better attention and executive function and quality of life at baseline. As this simple method to evaluate sleep quality and OSA is based on analyzing signals that are simple to collect, the method is practical for sleep-testing, over multiple nights and on multiple occasions. This method may assist physicians and parents to determine the most appropriate therapy for their child as some children may benefit from WWSC rather than interventions. If the parameters can be used to plan care a priori, this would provide a fundamental shift in how childhood OSA is diagnosed and managed.

## 1. Introduction

Sleep disordered breathing (SDB), characterized by abnormal respiratory and ventilation patterns during sleep, is a prevalent condition in children with disease severity ranging from primary habitual snoring (prevalence of 6–25%) to complete obstruction of the airway (estimated prevalence 3–6%) [1,2]. Obstructive sleep apnea (OSA) is diagnosed in children when obstructive apnea-hypopnea index (AHI) is ≥1.0 per hour of sleep [3].

The most common risk factors for OSA in children are enlarged tonsils and/or adenoids and obesity, but upper and lower airway disease, allergic rhinitis, low muscle tone and certain cranial structures such as midface deficiency and mandibular hypoplasia that may affect the size of the upper airway may also cause OSA [4]. Symptoms of OSA include mouth breathing, labored breathing, snoring, pauses in breathing, frequent night waking, nocturnal enuresis and the child sleeping in unusual positions (seated position, propped up or with the neck hyperextended). OSA may disrupt and shorten sleep and consequently may cause morning headache, daytime sleepiness and irritability that in children commonly presents as annoyance, behavioral and attention problems, and poor academic functioning [5,6]. In children though, there is often not a clear correlation between clinical symptoms and OSA severity. Furthermore, the literature indicates that the outcome of physical examinations such as the Mallampati score or the Friedman palate position, or subjective sleep questionnaires do not correlate well with severity of airway obstruction and may not be reliable when evaluating children for OSA [7,8,9]. Based on the results of the Childhood Adenotonsillectomy Study for Children with OSA (CHAT) the ability of otolaryngologists to correctly predict the presence of Polysomnography (PSG)-confirmed OSA based on clinical examination and history was only approximately 50% [10], there is a need for better ways to objectively identify children who have OSA and would benefit from therapy. 

The CHAT-study is currently the largest randomized controlled study in children with OSA without prolonged oxyhemoglobin desaturation. The study was rigorously conducted, evaluating multiple outcomes in a large cohort of otherwise healthy children 5 to 9 years old. The primary outcome was to evaluate if early adenotonsillectomy (eAT) better improved attention and executive function utilizing the Developmental Neuropsychological Assessment (NEPSY) when compared to watchful waiting with supportive care (WWSC). The premise of remission of OSA was based on improvement in AHI_Obstructive_ to be reduced to <1.0/h [10]. The results indicate that eAT when compared to WWSC did not result in significantly greater improvements in attention and executive function after a period of 7-months evaluated with NEPSY [10] but secondary outcome measures of AHI and OSA symptoms, behavior and quality of life were improved [11,12,13,14,15,16]. 

Adenotonsillectomy is a common surgical procedure and often the first line of therapy in children in the USA [17]. The finding in the CHAT-study that 46% of children in the WWSC-group did normalize their AHI to levels <1.0 when re-evaluated during the follow-up at 7 months [10] leaves the question, if the group of children that would likely show spontaneous resolution over time could be set apart from the group that would benefit most from Adenotonsillectomy [18] ? Physicians and parents need to carefully evaluate the risks and benefits of adenotonsillectomy. Surgery poses potential immediate complications for the child [19] and recent data suggested that removing this lymphoid tissue with surgery also may lead to increased risk for longer-term respiratory, allergic and infectious sequelae [20]. Moreover, residual disease following adenotonsillectomy is very common, with just over a quarter of the pediatric OSA patients achieving complete resolution of the disease post-surgery [21].

AHI is a commonly used metric to describe OSA and its severity, but with its limitation of being just an event-based frequency parameter, it may not sufficiently reflect the complex pathophysiological mechanisms of OSA and even more so in children than adults. Changes in autonomic function during sleep may provide an adequate and easier marker of OSA related sleep fragmentation in children as compared to electroencephalographical (EEG) arousals and offer additional information that may be of value when choosing the most appropriate therapy for children with OSA. 

To assess changes in sleep based on changes in autonomic function, data from the CHAT-study were analyzed, utilizing a cloud-base software technology, SleepImage^®^ System (MyCardio LLC, Denver, CO, USA). This method makes use of cardiopulmonary coupling (CPC) calculations to automatically generate sleep outcome metrics, including a sleep quality index (SQI) as an objective measure of overall sleep health, and AHI to identify OSA, when oximetry data are available [22,23,24]. The SQI is heavily weighted by the amount of stable sleep, which is a dimension relatively reflective of conventional sleep stages derived from traditional PSG, and strongly covaries with EEG slow-wave power. However, stable sleep and SQI are not limited by conventional EEG-slow wave sleep (N3) categorization. As continuous electrocardiographical (ECG) or photoplethysmographical (PPG) recordings are sufficient as an input signal for the CPC-analysis, the SQI can be computed from a comfortable finger-worn home sleep testing device (Figure 1) [25]. We hypothesized that SQI may provide additional information to identify those children with OSA that are more likely to have spontaneous remission of OSA and may benefit from WWSC and/or other amicable therapies with repeated sleep testing in the child’s own sleeping environment rather than eAT.

## 2. Methods

### 2.1. Study Design

The CHAT-study (NCT005600859) was a USA based multicenter, single-blinded, randomized controlled trial (RCT) designed to assess if children with OSA without prolonged oxyhemoglobin desaturations assigned to eAT would improve better in cognitive function than children assigned to WWSC [26]. The obstructive sleep apnea syndrome was defined as an obstructive apnea–hypopnea index (AHI) score of 2.0 or more events per hour (AHI: Obstructive apneas with ≥3% oxygen desaturation with arousal and hypopneas with >50% flow reduction and ≥3% oxygen desaturation with or without arousal divided by hours of total sleep time) or an obstructive apnea index (AHI_Obstructive_: Obstructive apneas with no oxygen desaturation threshold used and with or without arousal divided by total hours of sleep) score of 1.0 or more events per hour [10]. Institutional review board approval was obtained from each participating institution, children provided assent and parents (caretakers), provided written informed consent. A detailed description of the methodology [26] and results of the trial’s primary outcomes have been reported [10]. For this analysis, data use agreement was obtained from the National Sleep Research Resource [27]. 

### 2.2. Participants

Habitually snoring children, aged 5 to 9 years, were recruited from pediatric sleep centers, pediatric otolaryngology clinics, general pediatric clinics, and the general community. All children were referred to have a PSG-study at baseline, and if OSA was confirmed the child was randomized to early adenotonsillectomy (eAT) or watchful waiting with supportive care (WWSC) with a follow-up after 7 months. OSA was confirmed when AHI_Obstructive_ ≥ 1.0 events per hour. An otolaryngology evaluation also confirmed the child as a candidate before adenotonsillectomy when the tonsillar hypertrophy score ≥ 1 based on a standardized scale of 0–4. Study exclusion criteria included severe OSA (AHI_Obstructive_ > 20.0, AHI > 30.0, or a percentage of sleep time at an oxygen saturation of less than 90% for more than 2% of total sleep time), craniofacial or cardiac disorders, recurrent tonsillitis requiring surgical intervention, psychiatric or behavioral disorders (including attention-deficit/hyperactivity disorder) and extreme obesity (defined by a body mass index (BMI) z score > 2.99 for age group and sex) [10,26]. 

### 2.3. Group Assignment, Sleep-Study and Cognitive Assessments 

At baseline and before the child was randomized to either eAT or WWSC, the child completed a full-night polysomnography (PSG) sleep-study conducted by a study-certified technician using a standardized protocol following AASM guidelines. Scoring was performed according to AASM pediatric criteria by certified technologists blinded to all other study data at a central PSG reading center (Case Western Reserve University/Brigham and Women’s Hospital) [28]. During a separate visit cognition was evaluated; verbal skills and learning, nonverbal reasoning, attention and executive function, perceptual–motor and visual–spatial skills, by the developmental Neuropsychological Assessment (NEPSY) [29,30] was administered to each child individually as well as evaluation of behavior and quality of life. Testing was then readministered after 7-months to evaluate outcomes [26]. 

### 2.4. Cardiopulmonary Coupling Analysis

SleepImage is Health Insurance Portability and Accountability Act (HIPAA) compliant, Food and Drug Administration (FDA; K182618) cleared cloud-based software as a medical device (SaMD) [31] to establish sleep quality, evaluate sleep disorders and aid in diagnosis and management of SDB and OSA based on clinically validated algorithms using continuous ECG- or PPG-signal for CPC-analysis, collected with devices validated to have adequate sampling rates and signal quality (Figure 1). The software is based on calculations and spectral analysis of coupled interactions between two continuously collected physiological data-streams, both of which are strongly modulated by autonomic sleep regulating mechanisms (sympathovagal balance); ECG or PPG for heart/pulse rate variability (HRV/PRV) and respiration (ECG-derived respiration, EDR/PPG-derived respiration, PDR) [23,32]. The data are analyzed for the strength of the coupling between the two signals to look for the oscillation amplitude and coherence between the two signals at given frequency that allows for estimation of sleep stages. The software presents information on sleep duration, wake after sleep onset, sleep fragmentation and automatically calculates a summary output as a sleep quality index (SQI). The output is cleared to be fully automated, presented numerically and graphically in the sleep quality report (Figure 2) [23,24,33,34,35,36]. In this analysis of the CHAT-database the ECG-signal combined with oxygen saturation data (SpO_2_) was utilized for the CPC-analysis to calculate SQI and AHI to identify OSA [22,23,32].

In the CPC analysis, the spectral analysis output presents Non-Rapid Eye Movement (NREM) sleep as a distinct bimodal sleep structure of stable-sleep (high frequency coupling, HFC; 0.1–0.5 Hz) and unstable-sleep (low frequency coupling, LFC; 0.01–0.1 Hz) [25,32,37]. Stable-sleep is part of stage-NREM-2 and all of stage-NREM-3 sleep and is associated with non-cyclic alternating pattern (non-CAP) EEG, periods of stable breathing, increased delta power and blood pressure dipping. Conversely, unstable sleep is associated with sleep instability, characterized by variability of tidal volumes and non-dipping of blood pressure and CAP-EEG [25,33,34,38]. In a subset of the low-frequency coupling band described as elevated low frequency coupling (eLFC), periods of apneas and hypopneas exceeding clinical thresholds, sleep fragmentation and arousals (eLFC_BB_) as well as apneas caused by respiratory dyscontrol (periodicity, eLFC_NB_) are displayed [32,39]. Wake and Rapid Eye Movement (REM) sleep are presented as very low frequency Coupling (vLFC; 0–0.01 Hz), where REM has two subsets (bimodal), as Stable- and Unstable-REM sleep based on classification of the dominant state by the frequency analysis, where fragmented REM sleep is often accompanied by eLFC [38]. The output parameters sleep stability, fragmentation, and periodicity are reflected in the Sleep Quality Index (SQI), displayed on scale of 0 to 100, where a higher SQI reflects a better sleep. Stable-sleep and REM-sleep positively affect SQI while dominance of unstable sleep, excessive wake, cyclic variation of heart rate (CVHR) [40,41], sleep fragmentation and periodicity, negatively affect SQI [23,24,25,34,35,36,37,38,42]. When an SpO_2_ signal is recorded simultaneously with the ECG- or PPG-signal, AHI is calculated and reported. The sAHI is automatically calculated from CPC-analysis of the ECG- or PLETH- signal where non-hypoxic events are calculated based on the frequency at which elevated low-frequency coupling (eLFC_BB_ or eLFC_NB_) occurs. Hypoxic events are detected through the SpO_2_ data where a qualifying event is characterized by minimum of 10 s duration and 3% oxygen desaturation. Total number of events recorded is divided by the sleep period to generate an index value, sAHI [22,43]. 

### 2.5. Outcome Variables

The primary output is to compare (1) clinical characteristics including age, weight (BMI z-score waist circumference), evaluation of tonsillar size (Mallampati score), palate position (Friedman palate position), history of allergies and/or asthma; (2) symptoms of OSA (OSA-18, Pediatric Sleep Questionnaire-Sleep Related Breathing Disorders (PSQ-SRBD)), behavior (Child Behavior Checklist, Conners rating scale) and quality of life (Pediatric Quality of Life Questionnaire (PedQoL); (3) cognitive functioning (NESPY) and; (4) objective sleep measures, OSA severity (AHI) and sleep quality (SQI) at baseline in children that had spontaneous resolution of OSA and high sleep-quality when evaluated after 7-months of WWSC (Group-1_Remission_) to those that did not improve (Group-2_OSA_). 

### 2.6. Statistical Analysis

The primary analysis utilized analysis of covariance (ANCOVA), controlling for age, gender, and site, to compare the subjects stratified as having SQI ≥ 75 and AHI < 1.0 at follow-up (Group-1_Remission_) and those that did not (Group-2_OSA_). The baseline values are reported in Table 1 as margins with the associated 95% confidence interval, with the contrast between the groups presented with the associated *p*-value in parenthesis. Additionally, the delta values from baseline to follow-up are presented in Table 2 with the contrast (difference-in-difference) between the groups presented with the associated *p*-value. STATA version 15.1 (StataCorp LLC) was used for the analysis.

## 3. Results

ECG- and SpO_2_-signals were successfully analyzed from all the 203 subjects in the WWSC-group with data at both baseline and follow-up. At baseline Group-1_Remission_ (n = 43; 21%) were significantly more likely to have high SQI 79.96 [CI_95%_ 75.05, 84.86] vs. 72.44 [CI_95%_ 69.50, 75.39]; *p* = 0.005, mild OSA AHI_Obstructive_ 4.01 [CI_95%_ 2.34, 5.68] vs. 6.52 [CI_95%_ 5.47, 7.57], *p* = 0.005; lower BMI z-score 0.44 [CI_95%_ 0.05, 0.83] vs. 0.94 [0.69, 1.19], *p* = 0.018 and average waist circumference 58.34 [CI_95%_ 54.29, 62.38] vs. 62.74 [CI_95%_ 60.20, 65.29], *p* = 0.042; higher NEPSY-core domain attention and executive function scores 106.22 [CI_95%_ 101.67, 110.77] vs. 101.14 [CI_95%_ 98.58, 103.72], *p* = 0.037, NEPSY-Tower-scaled-scores 11.51 [CI_95%_ 10.64, 12.43] vs. 10.03, [CI_95%_ 9.46, 10.6], *p* = 0.008 and better quality of life according to parents 83.74 [CI_95%_ 78.95, 88.54] vs. 77.51 [CI_95%_ 74.49, 80.53], *p* = 0.015. There was no significant difference comparing the groups in Mallampati score, Friedman palate position, symptoms of OSA evaluated with OSA-18 or PSQ-SRBD or behavior evaluated with the Child Behavior Checklist and Conners DSM-IV.

## 4. Discussions

In this secondary analysis of the CHAT-study, children that had spontaneous remission of OSA and higher sleep quality at follow-up (Group-1_Remission_) showed better sleep quality at baseline, mild OSA and were likelier to be healthy weight. They also had better baseline measures of attention and executive function and quality of life than children with OSA and lower-sleep quality at follow-up (Group-2_OSA_). The groups did not differ when evaluated based on clinical observations, Mallampati Score or Friedman palate position or by utilizing questionnaires evaluating sleep related symptoms and behavior. With sleep and its correlated outcome metrics such as quality of life and better cognitive functioning in the attention and executive domain being key factors in the decision to decide on a therapy, the results indicate that this simple method to evaluate sleep may assist physicians and parents to determine the most appropriate therapy for their child because some children may benefit from watchful waiting rather than interventions. If the parameters can be used to plan care a priori, this would provide a fundamental shift in how childhood OSA is diagnosed and managed.

A previous report by Chervin and coworkers [44] from the CHAT-study found that 42% of children previously considered surgical candidates by their otolaryngologists, no longer met polysomnographic criteria for OSA at follow-up. In their analysis, stratifying the group strictly based on AHI improvement, and similar to our analysis, they identified that lower AHI_Obstructive_ and central obesity were independent predictors of disease remission and additionally that children that had remission of their OSA had less sleep related symptoms evaluated by the pediatric sleep questionnaire (PSQ-SRBS). When including objective sleep quality (SQI) in addition to AHI_Obstructive_ to define remission, 21% of children in the WWSC-group improved. These children had good sleep quality, mild OSA, were more likely to be in healthy weight, have better quality of life and cognitive function but did not differ in OSA-symptoms or overt behavior at baseline. During NREM-sleep, slow-wave activity (SWA), has been suggested as essential to learning and memory processes, and from the perspective of brain maturation [45]. Moreover, it is known that fragmented sleep or sleep deprivation negatively impacts daytime executive functioning that require the prefrontal cortex. Tasks that involve planning, monitoring and focusing, such as the Tower, are hence sensitive to the effects caused by poor sleep [46]. The SQI is relatively independent of conventional sleep stages but does covary with EEG slow-wave power and has a strong correlation to PSG-measured arousals (Hilmisson et al., 2019b, Thomas et al., 2014) [24,25]. Higher SQI, reflecting better, less fragmented sleep and more NREM-stable sleep at baseline may thus facilitate OSA having fewer negative effects on child’s cognitive performance and quality of life and these children may therefore possibly be less likely to improve these parameters with surgery. 

Obesity is a known risk factor for OSA in children and adults and these results demonstrate that spontaneous resolution of OSA is less likely in obese children. A recent study by Jacobs et al., evaluated the effect of weight loss in children 8–18 years old with obesity and OSA on endothelial function. The study reported 67% prevalence of OSA and improvement in endothelial function with weight loss. In children with more severe obesity despite weight loss, the likelihood of a residual OSA was increased [47].Given the increasing prevalence of obesity in children [48], it is imperative to have reliable and cost effective ways to screen for OSA and for disease management.

A recent study in healthy children suspected of OSA reported that 18% of children undergo surgery without objective sleep evaluation and that of the children evaluated with PSG-studies (based on parent’s concern and preference for their child to have objective sleep-evaluation before surgery) only about 55% had OSA and hence might benefit from surgery [49]. This is of concern as surgery includes some immediate risk for the child and may cause complications [19] and removing this lymphoid tissue may increase longer-term respiratory, allergic, and infectious sequelae [20]. Not only do a limited number of children get objective sleep testing before AT-surgery, but there is also even less evidence of follow-up and repeat PSG testing despite the high prevalence of persistent OSA in children according to a systematic review by Galluzzi et al., which included nine studies assessing for severe OSA post adenotonsillectomy. All the studies documented persistent OSA after adenotonsillectomy. The number of residual OSA considering AHI > 5.0 varied from 30.0 to 55.5%, in case of AHI > 1.0 from 60.0 to 90.6%. This highlights the lack of cure with adenotonsillectomy in children and the need for ongoing assessment postoperatively [50].

That almost half of the children (46%) in the CHAT study that did not undergo surgery normalized their PSG-findings (OSA) during the follow-up period based on AHI [10] may indicate that WWSC or other amicable therapies and multi-night sleep testing might be preferred before surgery is implemented in children with milder form of OSA and high SQI. Performing a surgery on a child without a need, may cause both unnecessary distress for the child and affect their future health prospects as well as incurring unnecessary cost for both parents and payers. Parents should have access to simple, low-cost, clinically valid, objective sleep measures in the process of choosing the best treatment options for their child as not all parents have the opportunity of a PSG-sleep study for their child and PSG is not a method that is easily implemented into every-day care nor is it repeatable or scalable.

Implementing a more longitudinal and comprehensive sleep care approach should improve clinical management of sleep disorders in children and contribute to the child’s future health prospects and quality of life. Objective sleep testing that can be repeated as appropriate, whether to optimize timing and selection of intervention or as a follow-up assessment after surgery as residual symptoms or persistent OSA are common should improve disease management. For this to be practical, solutions need to be clinically validated, simple to use for the child, their parents and physicians and suitable for repeated testing in the child’s own sleeping environment as well as low-cost to be scalable. The method evaluated in this manuscript offers the opportunity for both pre- and post-therapy assessment for all children with suspected sleep disorders based on its simplicity and low-cost. It also minimizes the first night effect of sympathetic stimulation caused by PSG-testing, which has previously been identified as a challenging environment for children [51].

### Strengths and Limitations

A strength of the CHAT-study is that the study was well-designed and rigorously conducted evaluating multiple outcomes in a large cohort of otherwise healthy children in the age range of 5 to 9 years across several academic centers across the United States. Limitations of the CHAT-study are that; (1) the children in the study had little hypoxemia because it was clinically unethical to subject more severe children to watchful waiting; (2) the narrow age range of children 5–9 years, as the peak growth in the lymphoid issue is during the preschool years (2–5 years of age) increasing the likelihood of symptoms of oSDB to develop at younger age and OSA affecting children from 2 to 9 years of age; (3) having only one night of sleep data there is no option to evaluate night to night variability and; (4) the short follow-up period may also have affected outcomes. Limitations of this report are: (1) that it is a secondary analysis of the reported study to try and determine other factors that may be predictors of resolution of disease with watchful waiting, and; (2) we applied the approach of simplicity and for further research including other CPC-biomarkers in the hypothesis we should give a more comprehensive information in relation to changes in neurocognition.

## 5. Conclusions

This study evaluated clinical characteristics and symptoms in children likely to have spontaneous remission of OSA with WWSC based on sleep quality, obstructive breathing, quality of life and cognitive function. Children that spontaneously improved their OSA had objectively measured higher sleep quality, mild OSA, were more likely to be in healthy weight and had better cognition but did not differ when evaluated by clinical parameters of Mallampati Score, Friedman palate position or by questionnaires evaluating sleep related symptoms.

## Figures and Tables

**Figure 1 children-08-00980-f001:**
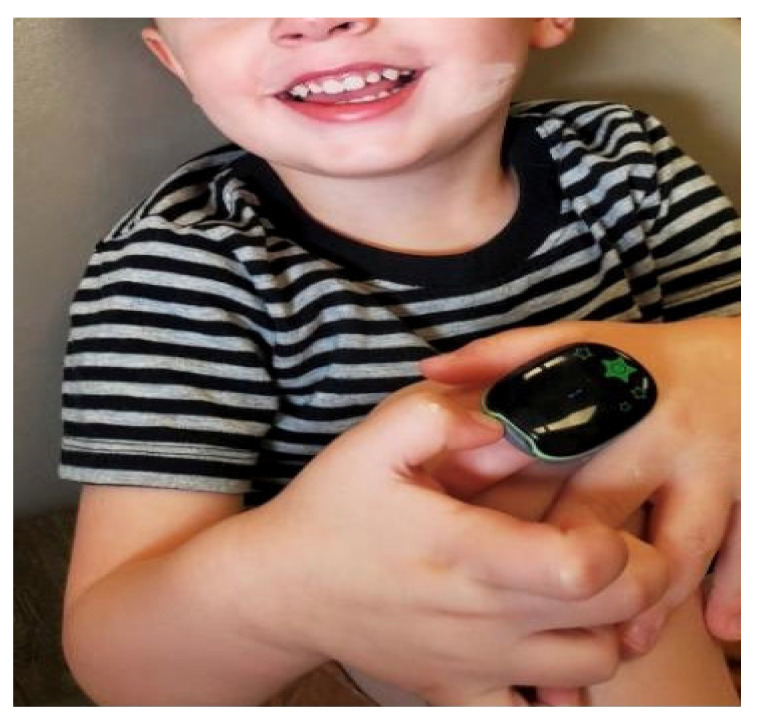
The SleepImage finger-worn sleep recorder.

**Figure 2 children-08-00980-f002:**
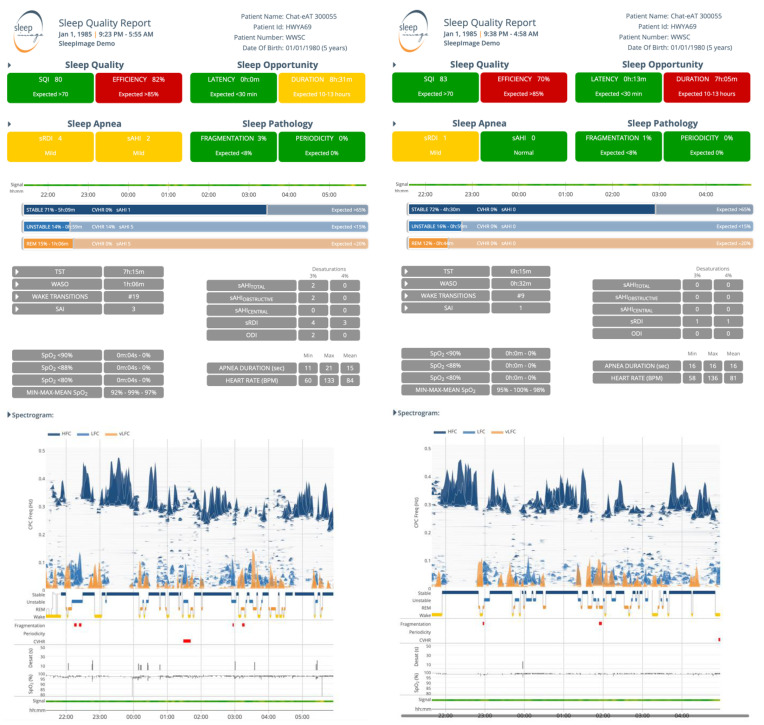
Sleep Quality Reports from a sleep study of a 5-year-old child randomized to Watchful Waiting with Supportive Care (WWSC) in the CHAT study at baseline and 7-month follow-up.

**Table 1 children-08-00980-t001:** Baseline characteristics of children randomized to watchful waiting with supportive care (WWSC) classified based on resolution of their sleep apnea (AHI < 1) and good sleep quality (SQI ≥ 75) at 7-month follow-up.

	Higher Sleep Quality and No OSA at Follow-Up (n = 43)	Lower Sleep Quality and OSA at Follow-Up (n = 160)	Contrast (*p*-Value)
**Characteristics**			
Male (%)	0.63 [0.45, 0.80]	0.53 [0.42, 0.63]	0.10 (0.234)
African American (%)	0.38 [0.22, 0.53]	0.45 [0.35, 0.54]	−0.07 (0.413)
Caucasian (%)	0.40 [0.24, 0.55]	0.38 [0.28, 0.47]	0.02 (0.826)
Other (%)	0.23 [0.14, 0.32]	0.18 [0.12, 0.23]	0.05 (0.291)
Age (years)	6.33 [5.87, 6.79]	6.46 [6.17, 6.75]	−0.13 (0.590)
Body Mass Index (BMI) (Kg/m^2^)	17.40 [15.90, 18.89]	19.10 [18.16, 20.05]	−1.71 (0.033)
BMI-z score	0.44 [0.05, 0.83]	0.94 [0.69, 1.19]	−0.50 (0.018)
Waist circumference—Average (cm)	58.34 [54.29, 62.38]	62.74 [60.20, 65.29]	−4.40 (0.042)
Healthy weight	0.72 [0.56, 0.88]	0.54 [0.44, 0.64]	0.18 (0.035)
Mallampati Score I and II	0.65 [0.49, 0.81]	0.58 [0.48, 0.68]	0.07 (0.405)
Friedman palate position I and II	0.48 [0.33, 0.54]	0.37 [0.28, 0.47]	0.11 (0.182)
Allergies	0.30 [0.14, 0.45]	0.30 [0.20, 0.39]	0.00 (0.995)
Asthma	0.25 [0.10, 0.40]	0.27 [0.18, 0.37]	−0.02 (0.806)
**Sleep Measures**			
Sleep Quality Index (SQI)	79.96 [75.07, 84.86]	72.44 [69.50, 75.39]	7.52 (0.005)
Apnea hypopnea index 3% (AHI)	4.01 [2.34, 5.68]	6.52 [5.47, 7.57]	−2.51 (0.005)
Respiratory Disturbance Index 3% (RDI)	5.39 [3.66, 7.12]	8.13 [7.04, 9.22]	−2.74 (0.003)
**Questionnaires**			
PedQoL Parent Total Score	83.74 [78.95, 88.54]	77.51 [74.49, 80.53]	6.23 (0.015)
PedQoL Child Total Score	70.84 [65.91, 75.77]	67.59 [64.49, 70.68]	3.25 (0.217)
Child Behavior Checklist Total score	51.51 [47.91, 55.10]	53.73 [51.50, 55.97]	−2.23 (0.244)
Conners DSM-IV Total Score	52.21 [48.54, 55.87]	52.66 [50.26, 56.13]	−1.61 (0.408)
OSA-18 Summary-Average	2.78 [2.43, 3.13]	3.12 [2.90, 3.34]	−0.34 (0.067)
PSQ-SRBD Total	0.44 [0.38, 0.50]	0.48 [0.44, 0.51]	0.04 [0.222]
**NEPSY**			
Core Domain: Attention/Executive function	106.22 [101.67, 110.77]	101.14 [98.56, 103.72]	5.08 (0.037)
Tower: Attention/Executive function	11.54 [10.64, 12.43]	10.03 [9.46, 10.6]	1.27 (0.008)

Descriptive statistics were presented as means with confidence intervals (CI_95%_). BMI, body mass index (the weight in kilograms divided by the square of the height in meters); BMI z score (relative weight adjusted for child age and sex); SQI, sleep quality index; AHI, apnea/hypopnea index; RDI, respiratory disturbance index; PedQoL, Pediatric Quality of Life questionnaire; PSQ-SRBD, Pediatric Sleep Questionnaire-Sleep Related Breathing Disorders; NEPSY, Developmental Neuropsychological Assessment.

**Table 2 children-08-00980-t002:** Changes in characteristics of children randomized to watchful waiting with supportive care (WWSC) during the study period when classified based on resolution of their sleep apnea (AHI < 1) and sleep quality (SQI ≥ 75) at 7-month follow-up.

	Higher Sleep Quality and No OSA at Follow-Up (n = 43)	Lower Sleep Quality and OSA at Follow-Up (n = 160)	Contrast (*p*-Value)
**Characteristics**			
Body Mass Index (BMI) (Kg/m^2^)	0.48 [0.13, 0.82]	0.53 [0.31, 0.75]	−0.06 (0.763)
BMI-z score	−0.01 [−0.33, 0.31]	−0.23 [−0.43, −0.03]	0.22 (0.204)
Waist circumference—Average (cm)	1.48 [0.18, 2.79]	2.37 [1.53, 3.20]	−0.88 (0.203)
**Sleep Measures**			
Sleep Quality Index (SQI)	6.31 [−0.05, 12.67]	−1.65 [−5.59, 2.29]	7.96 (0.023)
Apnea hypopnea index 3% (AHI)	−3.08 [−5.79, −0.38]	0.05 [−1.65, 1.76]	−3.14 (0.030)
Respiratory Disturbance Index 3% (RDI)	-3.08 [−5.95, −0.22]	0.26 [−1.55, 2.06]	−3.34 (0.029)
**Questionnaires**			
PedQoL Parent Total Score	−4.53 [−8.66, −0.39]	−1.58 [−4.18, 1.03]	−2.95 (0.182)
PedQoL Child Total Score	0.78 [−5.05, 6.61]	1.63 [−2.37, 5.62]	−0.85 (0.781)
Child Behavior Checklist Total score	−1.17 [−3.84, 1.50]	−1.41 [−3.07, 0.26]	0.24 (0.868)
Conners DSM-IV Total Score	−0.18 [−3.21, 2.85]	−0.38 [−2.30, 1.55]	0.20 (0.902)
OSA-18 Summary-Average	−0.11 [−0.48, 0.26]	−0.16 [−0.39, 0.07]	0.05 (0.793)
PSQ-SRBD Total	−0.03 [−0.09, 0.03]	0.001 [−0.03, 0.05]	−0.04 (0.215)
**NEPSY**			
Core Domain: Attention/Executive function	0.72 [−3.6, 5.03]	3.45 [0.73, 6.16]	−2.73 (0.235)
Tower: Attention/Executive function	−0.09 [−1.06, 0.87]	0.53 [-0.07, 1.14]	−0.63 (0.233)

Descriptive statistics were presented as means with confidence intervals (CI_95%_). BMI, body mass index (the weight in kilograms divided by the square of the height in meters); BMI z score (relative weight adjusted for child age and sex); SQI, sleep quality index; AHI, apnea/hypopnea index; RDI, respiratory disturbance index; PedQoL, pediatric quality of life questionnaire; PSQ-SRBD, Pediatric Sleep Questionnaire-Sleep Related Breathing Disorders; NEPSY, Developmental Neuropsychological Assessment.

## Data Availability

The data set used was provided by the National Sleep Research Resource, with access to the CHAT-database http://sleepdata.org/dataseets/chat (accessed on 21 October 2021). The National Institutes of Health, NHLBI, supported the CHAT-research: R24Hl114473.

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
