# Peer review of "Characteristics of Children Likely to Have Spontaneous Resolution of Obstructive Sleep Apnea: Results from the Childhood Adenotonsillectomy Trial (CHAT)"

_children, 2021, doi:10.3390/children8110980_

Round 1

Reviewer 1 Report

Study strengths:

-Clinically very important topic. This is an extremely important topic and something that pediatric sleep clinicians deal with on day to day basis.

-Interesting that the SleepImage software was used, since this is already being used by families out in the community. 

Suggestions to strengthen manuscript:

-The authors should take care in how they define 'normalization' of the obstructive AHI in the context of the CHAT study. While I agree that a technically 'normal' OAHI is <1/hr, the CHAT study defined normalization as AHI <2/hr AND OAI<1/hr. The OAI and OAHI are different. Please clarify in the text.

-Typo in Figure 2 legend. "ranomized"

-Provide a little description of how AHI is derived from the PPG signal of the SleepImage software.

-Were both the ECG and PPG signals used by the SleepImage software to derive metrics? Is this the same as the commercially available device?

-I would suggest analyzing the data a little differently. Tables 1 and 2 are confusing. The more interesting clinical questions in my mind are: do the alternative metrics derived from the SleepImage software better correlate with baseline neuropsych metrics and quality of life data? Are changes in the novel metrics from the SleepImage software from baseline to followup better correlate with changes in those neuropsych/QOL outcomes over time? Are those correlations better than the correlation from our traditional metrics (AHI, O2 nadir, etc)? That data is in the CHAT and should be able to be directly compared to the SleepImage metrics.

-Do children who have "resolved" OSA by our conventional AHI metric but remain symptomatic have different SleepImage scores compared to those who have "resolved" OSA by our conventional AHI but also resolved symptoms?

-Was AHI derived from the device (PPG signal) or from the PSG itself? If it was derived from the device, can the authors compare the AHI from those two methods (SleepImage calculated vs PSG calculated)?

-Overall, the big question in my mind that this study should be able to answer (and I would reframe the analysis to address this question) is: Does this novel device (with its novel metrics) better characterize OSA-associated disease burden compared with traditional PSG-metrics (ie supplant the AHI, which is a flawed metric)? If so, this is of tremendous clinical value.

Author Response

Dear Reviewer, #1.

We appreciate your time reviewing our manuscript and value your comments and feedback.

-The authors should take care in how they define 'normalization' of the obstructive AHI in the context of the CHAT study. While I agree that a technically 'normal' OAHI is <1/hr, the CHAT study defined normalization as AHI <2/hr AND OAI<1/hr. The OAI and OAHI are different. Please clarify in the text.

This has been addressed and the text now states:

The obstructive sleep apnea syndrome was defined as an obstructive apnea–hypopnea index score of 2 or more events per hour (AHI: Obstructive apneas with ³3% oxygen desaturation with or without arousal and hypopneas with >30% flow reduction and ³3% oxygen desaturation with or without arousal) or an obstructive apnea index (AHIObstructive: Obstructive apneas with no oxygen desaturation threshold used and with or without arousal/hour of sleep) score of 1 or more events per hour.”

Typo in Figure 2 legend. "ranomized"

This has been addressed.

-Provide a little description of how AHI is derived from the PPG signal of the SleepImage software.

A description has been added to the Cardiopulmonary Coupling analysis part of the method section:

The sAHI is automatically calculated from CPC-analysis of the ECG- or PLETH- signal where non-hypoxic events are calculated based on the frequency at which elevated low-frequency coupling (eLFCBB or eLFCNB) occurs. Hypoxic events are detected through the SpO2 data where a qualifying event is characterized by minimum of 10 second duration and 3% oxygen desaturation. Total number of events recorded is divided by the sleep period to generate an index value, sAHI.”

Were both the ECG and PPG signals used by the SleepImage software to derive metrics? Is this the same as the commercially available device?

The data was analyzed using the ECG and SpO2 signals from the CHAT database, using the FDA-cleared and commercially available SleepImage System, which is Software as a Medical Device that uses the same algorithms to analyze ECG and PLETH signals. The SpO2 signal is analyzed independently of the ECG or PLETH analysis as outlined in the FDA clearance letter Indications for Use:

The SleepImage System is Software as a Medical Device (SaMD) that establishes sleep quality. The Sleepimage System analyzes, displays, and summarizes Electrocardiogram (ECG) or Plethysmogram (PLETH) data, typically collected during sleep, that is intended for use by or on the order of a Healthcare Professional to aid in the evaluation of sleep disorders, where it may inform or drive clinical management for children, adolescents, and adults. Sleepimage Apnea Hypopnea Index (sAHI), presented when oximeter data is available, is intended to aid healthcare professionals in diagnosis and management of sleep disordered breathing.”

-I would suggest analyzing the data a little differently. Tables 1 and 2 are confusing. The more interesting clinical questions in my mind are: do the alternative metrics derived from the SleepImage software better correlate with baseline neuropsych metrics and quality of life data? Are changes in the novel metrics from the SleepImage software from baseline to followup better correlate with changes in those neuropsych/QOL outcomes over time? Are those correlations better than the correlation from our traditional metrics (AHI, O2 nadir, etc)? That data is in the CHAT and should be able to be directly compared to the SleepImage metrics. Do children who have "resolved" OSA by our conventional AHI metric but remain symptomatic have different SleepImage scores compared to those who have "resolved" OSA by our conventional AHI but also resolved symptoms?

We agree that further analysis of the data will derive additional interesting information. This manuscript is based on the hypothesis that the SQI may provide information additional to the AHI to identify children with OSA who are more likely to have spontaneous remission and may therefore possibly benefit from WWSC and/or other amicable therapies with repeated sleep testing. We highly appreciate your suggestions that we find worthy of another manuscript submission in addition to this manuscript. To address this further we have added the following as a limitation: we applied the approach of simplicity and for further research including other CPC-biomarkers in the hypothesis should give a more comprehensive information in relation to changes in neurocognition.

-Was AHI derived from the device (PPG signal) or from the PSG itself? If it was derived from the device, can the authors compare the AHI from those two methods (SleepImage calculated vs PSG calculated)?

In this study the AHI is derived from both the ECG-signal (CPC-analysis) and the PPG signal for SpO2 data collected by the PSG systems used in the CHAT study. The SleepImage calculation of the AHI vs. the scoring of the PSG-derived AHI has already been published and is referenced in the manuscript; Hilmisson, H., et al. Sleep apnea diagnosis in children using software-generated apnea-hypopnea index (ahi) derived from data recorded with a single photoplethysmogram sensor (PPG); Results from the childhood adenotonsillectomy study (chat) based on cardiopulmonary coupling analysis. Sleep Breath, 24 (4), 1739-1749.

-Overall, the big question in my mind that this study should be able to answer (and I would reframe the analysis to address this question) is: Does this novel device (with its novel metrics) better characterize OSA-associated disease burden compared with traditional PSG-metrics (ie supplant the AHI, which is a flawed metric)? If so, this is of tremendous clinical value.

We believe that this question is partially answered in this manuscript demonstrating the additional value of the SQI when read in addition to the AHI to improve precision and timing of care as stated in our response above. We agree that there is more to be learned about how metrics in addition to the AHI can contribute to precision of care in the diagnostic and more importantly in the treatment process of children with sleep disorders. We will continue this work with the aim to keep our manuscripts focused to increase the likelihood of reaching pediatricians who are not trained in sleep medicine.

We thank you kindly for your review of this submission.

On behalf of the authors, Solveig.

Reviewer 2 Report

The paper " Characteristics of Children likely to have Spontaneous Resolu- 2
tion of Obstructive Sleep Apnea: results from the Childhood 3
Adenotonsillectomy Trial (CHAT)" by Magnusdottir et is an evaluation of the largest randomized controlled study in children with OSA without prolonged O2 desaturation. The authors propose a method that may assist physicians to select children likely to have spontaneous remission of OSA. There are minor aspects that can be answered.

  1. Page 4, line 138. Why did you choose the follow-up at 7 months ?
  2. Page 8, discussion. Do you have data about the medication during study? 
  3. Page 8, discussion. Can you suggest the most best indication for surgery? 

Author Response

Dear Reviewer, #2.

We appreciate your time reviewing our manuscript and value your comments and feedback.

Page 4, line 138. Why did you choose the follow-up at 7 months?

We did not choose the timing of the follow-up at 7 months; this was how the CHAT-study was set up. Your comment does though point to one of the criticisms of the CHAT-study having a to short follow-up period to identify changes in neurocognitive function.

Page 8, discussion. Do you have data about the medication during study? 

There is limited information about medication the children were taking during the study, limited to if the child was taking medication for e.g., asthma, ADHD, cancer, reflux, depression etc. but not type of medication.

Page 8, discussion. Can you suggest the most best indication for surgery? 

Based on the limited result of this analysis, one may want to initiate WWSC in healthy weight children with good sleep quality and mild OSA, with or without other therapies that have been suggested in this patient population like nasal steroids or Montelukast and include regular therapy tracking. This analysis does not address what would be the most or best indication for surgery.

Round 2

Reviewer 1 Report

I thank the authors for revising their manuscript based on my comments. A couple points:

  1. Please make sure that the description and abbreviations regarding AHI, obstructive AHI, and OAI are used appropriately and in line with how they were used in the CHAT study (I provided from their methods below). obstructive AHI is NOT equivalent to OAI.

"The AHI was defined as the sum of all obstructive and mixed apneas, plus hypopneas associated with a 50%

reduction in airflow and either a > 3% desaturation or electroencephalographic arousal, divided

by hours of total sleep time; the OAI as all obstructive apneas per sleep hour"

Author Response

The text in the manuscript has been adjusted per your suggestions: " The obstructive sleep apnea syndrome was defined as an obstructive apnea–hypopnea index (AHI) score of 2.0 or more events per hour (AHI: Obstructive apneas with ³3% oxygen desaturation with arousal and hypopneas with > 50% flow reduction and ³3% oxygen desaturation with or without arousal divided by hours of total sleep time) or an obstructive apnea index (AHIObstructive: Obstructive apneas with no oxygen desaturation threshold used and with or without arousal divided by total hours of sleep) score of 1.0 or more events per hour.